# Ultra-thin lithium aluminate spinel ferrite films with perpendicular magnetic anisotropy and low damping

Xin Yu Zheng [1,2,11] ✉, Sanyum Channa[2,3,11], Lauren J. Riddiford [1,2], Jacob J. Wisser[4], Krishnamurthy Mahalingam[5], Cynthia T. Bowers[5], Michael E. McConney[5], Alpha T. N'Diaye [6], Arturas Vailionis [7,8], Egecan Cogulu[9], Haowen Ren [9], Zbigniew Galazka[10], Andrew D. Kent [9] & Yuri Suzuki [1,2]

Ultra-thin films of low damping ferromagnetic insulators with perpendicular magnetic anisotropy have been identified as critical to advancing spin-based electronics by significantly reducing the threshold for current-induced magnetization switching while enabling new types of hybrid structures or devices. Here, we have developed a new class of ultra-thin spinel structure $Li_{0.5}Al_{1.0}Fe_{1.5}O_4$ (LAFO) films on $MgGa_2O_4$ (MGO) substrates with: 1) perpendicular magnetic anisotropy; 2) low magnetic damping and 3) the absence of degraded or magnetic dead layers. These films have been integrated with epitaxial Pt spin source layers to demonstrate record low magnetization switching currents and high spin-orbit torque efficiencies. These LAFO films on MGO thus combine all of the desirable properties of ferromagnetic insulators with perpendicular magnetic anisotropy, opening new possibilities for spin based electronics.

Ultra-thin ferromagnetic insulators (FMI) with perpendicular magnetic anisotropy (PMA) and low damping provide new opportunities for inducing emergent magnetic and topological phenomena at interfaces and efficiently sourcing, controlling and detecting pure spin currents, thereby changing the landscape of spin wave devices. FMIs with PMA have been shown to stabilize topological defects in the form of skyrmions that are robust to perturbations, and have been predicted to give rise to the quantum anomalous Hall effect when interfaced with topological insulators[1-3]. They also support the manipulation and isotropic propagation of spin waves in the absence of dissipative charge currents, providing a new paradigm for energy efficient spin-based computing and memory[4-9]. However, crucial to the success of PMA FMIs in applications is the presence of two additional features: low magnetic damping in the ultra-thin regime and high interface quality with adjacent spin-to-charge conversion layers. These factors optimize performance by decreasing switching current in spin-transfer-torque MRAM (STT-MRAM)[10], increasing domain wall velocity in racetrack memory[11] or increasing Dzyaloshinskii-Moriya interaction (DMI) strength to aid formation of skyrmions[12], among other effects.

[1]Department of Applied Physics, Stanford University, Stanford, CA 94305, USA. [2]Geballe Laboratory for Advanced Materials, Stanford University, Stanford, CA 94305, USA. [3]Department of Physics, Stanford University, Stanford, CA, USA. [4]National Institute of Standards and Technology, Gaithersburg, MD 20899, USA. [5]Air Force Research Laboratory, Wright Patterson Air Force Base, Dayton, OH 05433, USA. [6]Advanced Light Source, Lawrence Berkeley National Laboratory, Berkeley, CA 94720, USA. [7]Stanford Nano Shared Facilities, Stanford University, Stanford, CA 94305, USA. [8]Department of Physics, Kaunas University of Technology, Studentu Street 50, LT-51368 Kaunas, Lithuania. [9]Center for Quantum Phenomena, Department of Physics, New York University, New York, NY 10003, USA. [10]Leibniz-Institut für Kristallzüchtung, Max-Born-Str. 2, 12489 Berlin, Germany. [11]These authors contributed equally: Xin Yu Zheng, Sanyum Channa. ✉e-mail: xinzheng@stanford.edu

Previous reports of FMI thin film systems that possess both PMA and low damping were largely devoted to garnet structure systems[5,13–15]. Although garnets are well known for their low damping, many studies have shown the presence of a significant magnetic dead layer at the film-substrate interface due to interdiffusion, likely a result of their high growth temperatures of 600–900 °C[16–19]. However some studies have shown that a sharp interface with bulk magnetization properties can be obtained in garnet films[15,20,21]. The interdiffusion layer places an undesirable lower bound on the thickness of the magnetic layer, which when combined with the complex crystal structure of the garnets makes them difficult to integrate into heterostructures for applications. More importantly, Pt, the heavy metal (HM) of choice in FMI-based spintronics studies, grows amorphously or incoherently on these FMIs, impacting the efficient transfer of spin across the interface.

In search of an FMI that exhibits PMA and low damping at extremely low thicknesses while having a low thermal budget and high quality interfaces with HMs, we have realized a new class of ultra-thin low loss spinel structure $Li_{0.5}Al_{1.0}Fe_{1.5}O_4$ (LAFO) thin films. Structural characterization demonstrates a highly crystalline and defect-free film. When grown on $MgGa_2O_4$ (MGO) substrates, LAFO demonstrates strain-induced PMA and ultra-low magnetic damping as low as $\alpha = 6 \times 10^{-4}$ for 15 nm thick films and on the order of values reported in yttrium iron garnet (YIG) systems with PMA at room temperature[5,14,22]. Bilayers of LAFO and Pt exhibit efficient transfer of spin current from the HM to the FMI and have been attributed to the high quality of the Pt/LAFO interface. Spin-orbit torque (SOT) switching in Pt/LAFO is demonstrated with critical switching currents as low as $6 \times 10^5$ A/cm². Note that this value is for a FMI system and is an order of magnitude lower than typical values of $10^7$ A/cm² typically observed in garnet/Pt systems at similar fields[5,23,24]. We also estimate the spin torque efficiency using harmonic Hall measurements and extract large damping-like spin torque efficiencies. The combination of low damping, PMA, absence of magnetic dead layers, epitaxial Pt overlayers and low current density SOT switching in LAFO films demonstrates a new class of magnetic insulating thin film materials for spin wave-based spintronics. To this end, LAFO has already been incorporated in building hybrid spin-Hall nano-oscillators which are essential in accelerating spintronic applications[25].

## Results

### Structural characterization

LAFO films were grown by pulsed laser deposition on (001)-oriented $MgGa_2O_4$ (MGO) substrates (see Methods). Structural characterization of LAFO films indicates excellent epitaxy and crystallinity. Figure 1a shows an atomic resolution high-angle annular dark-field scanning transmission electron microscopy (HAADF-TEM) image of the microstructure along the [110] direction with no evidence of defects or interfacial layers which is corroborated by pronounced Kiessig fringes in X-ray reflectivity spectra (Supplementary Fig. S2). Figure 1b shows symmetric $2\theta - \omega$ X-ray diffraction (XRD) scans around the (004) peak of 15.1 nm and 4.1 nm thick LAFO films. Clear Laue oscillations can be seen around the 15.1 nm (004) film peak indicating coherent diffraction, whereas the 4.1 nm film is too thin to show Laue oscillations due to the substantial peak broadening associated with the finite film thickness. The film peak is shifted to higher angles compared to that of bulk LAFO indicating the reduction of the c-axis lattice constant. Note that the broadening of the 4.1 nm film peak is not due to poor film quality, but rather due to the lower thickness. Reciprocal space map of the ($\bar{1}\bar{1}5$) LAFO and MGO peaks in Fig. 1c further indicates the coherence of the in-plane film and substrate lattice parameters. Note that the RSM is taken on an asymmetric ($\bar{1}\bar{1}5$) peak with an out-of-plane component in order to capture information for both the in-plane and out-of-plane reciprocal vectors. Together, these results confirm epitaxial and dislocation-free growth of the LAFO film under coherent tensile

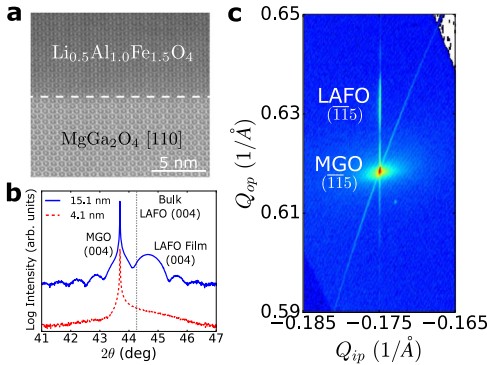

**Fig. 1 | Structural characterization of LAFO films on MGO. a** Atomic resolution high-angle annular dark-field scanning transmission electron microscopy (HAADF-TEM) image of the overall structure along [110], showing a clear film-substrate interface (dashed line). The difference in shading between the substrate and film is due to atomic Z contrast. **b** Symmetric $2\theta - \omega$ scan on the (004) peak, with the vertical dotted black line marking the bulk LAFO (004) peak position. **c** Reciprocal space map of the ($\bar{1}\bar{1}5$) peaks for the 15.1 nm film, showing alignment of in-plane wave-vector $Q_{ip}$ between the film and substrate peak.

strain on the MGO substrate (see Supplementary Material Section 1). Our results in the remainder of the manuscript will be focused on the 15.1 nm and 4.1 nm films. Magnetic characterization will be presented for the thicker film due to its cleaner signal, but SOT experiments will be performed on the thinner film due to its weaker anisotropy and ease of switching.

Pt/LAFO heterostructures show an epitaxial relationship between Pt and LAFO. We note that epitaxy does not mean single crystalline or in-plane aligned but merely the registry of the Pt layer with the underlying LAFO layer. This epitaxy differentiates the Pt/LAFO system from other Pt/FMI systems such as Pt/garnet bilayers. In Fig. 2a we show high-resolution TEM of the Pt/LAFO interface in which the transition between LAFO and Pt occurs within a monolayer. This smooth interface is further corroborated by the pronounced Kiessig fringes in X-ray reflectivity (Supplementary Material Section 1). The presence of Pt epitaxy is indicated by a prominent Pt (111) peak as seen in symmetric XRD scans in Fig. 2b. In-plane XRD scans shown in Fig. S2c reveal a complex epitaxial relationship between the Pt and LAFO involving a twinning pattern of the Pt domains, which is diagrammatically represented in Fig. S2d. Due to the three-fold symmetry of Pt [111] out-of-plane oriented unit cell and four-fold symmetry of LAFO/MGO [001] out-of-plane oriented unit cell, the Pt layer exhibit four twins that are rotated in-plane by 30 degrees from each other. This texturing manifests as 12 distinct in-plane peaks shown in (c). This epitaxial, but not in-plane aligned, growth of Pt on LAFO is in contrast to its incoherent growth on other materials[26,27]. The high quality interface facilitates the efficient transfer of spin current from the Pt to the FMI.

### Magnetic characterization

We performed SQUID magnetometry on LAFO films using an Evercool MPMS by Quantum Design at room temperature. We measure the magnetization as a function of field along the in-plane and out-of-plane directions. These measurements show that LAFO films exhibit PMA with bulk saturation magnetization values $M_s \approx 75$ kA/m at room temperature. Figure 3a shows magnetization (M) as a function of external magnetic field (H) on the 15.1 nm film. The magnetic easy axis lies out-of-plane along the [001] direction with a low coercivity of 1.6 mT and 100% remanence as shown in Fig. 3b. Conversely the in-plane [100] axis is magnetically hard, requiring a field of about 0.5 T for saturation. The small opening around zero field of the in-plane trace is due to a small misalignment of the sample with respect to the in-plane magnetic field. The origin of the PMA is due to magnetoelastic coupling as a result of the epitaxial bi-axial tensile strain on the film imposed by the substrate

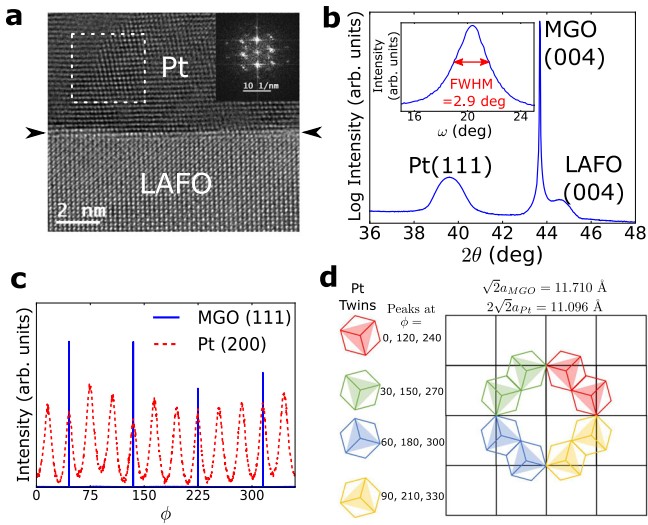

**Fig. 2 | Structural characterization of Pt/LAFO/MGO. a** Aberration corrected HRTEM image showing the Pt/LAFO interface. The image was obtain with a slight defocus (−10 nm) to delineate the interface (indicated by arrows), showing that transition between layers occurs within a monolayer. Inset: digital fast Fourier transform showing the orientation of the grain in the boxed region. **b** Symmetric $2\theta − \omega$ scan of a Pt(15 nm)/LAFO(15 nm)/MGO sample, showing the epitaxial Pt (111) peak. Inset: rocking curve on the Pt (111) peak. **c** In-plane X-ray diffraction scan of a Pt(15 nm)/LAFO(4 nm)/MGO sample on the MGO/LAFO (111) peaks and Pt (200) peaks, with $\phi$ as the azimuthal angle. **d** Epitaxial relationship between LAFO(001) and Pt(111) lattices.

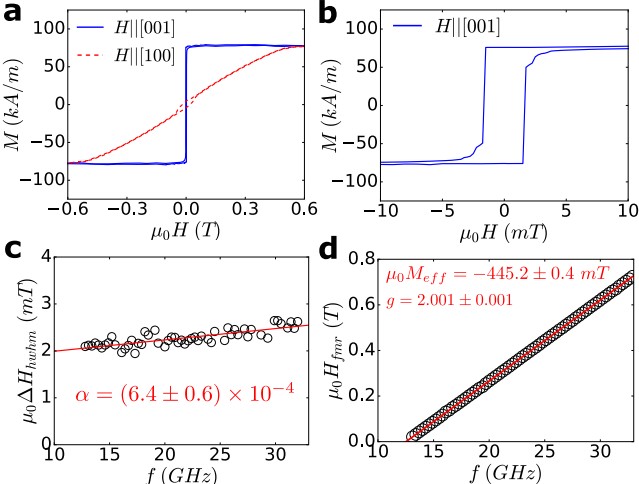

**Fig. 3 | Magnetic properties of the 15.1 nm LAFO film at room temperature. a** $M$ vs $H$ on the 15.1 nm LAFO film along the [001] and [100] axes. **b** Close up of the [001] trace near the origin. **c** FMR linewidth $\mu_0\Delta H_{hwhm}$ along the [001] axis as a function of frequency, with the red line as a linear fit. **d** FMR resonance field as a function of frequency, with the red line as a linear fit.

(see Supplementary Material Section 3)[22,28,29]. We show only the data for the 15.1 nm film here, but the $M_s$ and PMA are maintained across a range of LAFO thicknesses (see Supplementary Material Section 2). These results are consistent with the absence of a magnetic dead layer in the TEM data (Fig. 1a) present in other FMI thin film systems[16,30].

In terms of dynamic magnetic properties, LAFO films exhibit extremely low damping values. To characterize the damping, we perform room temperature broadband ferromagnetic resonance (FMR) in the out-of-plane direction. We fit each FMR spectra using a Lorentzian derivative lineshape, from which we obtain the FMR linewidth $\mu_0\Delta H_{hwhm}$ and FMR resonance field $\mu_0 H_{fmr}$. We then study the dependence of the linewidth and FMR resonance field as a function of the microwave frequency $f$ as shown in Fig. 3c and d. The analysis is described in Supplemental Material Section 3, from which we can extract the Gilbert damping parameter $\alpha = (6.4 \pm 0.6) \times 10^{-4}$ and an inhomogeneous broadening of $\mu_0\Delta H_0 = 1.5 \pm 0.1$ mT for the 15.1 nm film. This value of $\alpha$ is the lowest reported to date for spinel structure FMI films and is approaching those reported in YIG with PMA[5,14,22]. The inhomogeneous broadening is also similar to those observed in PMA YIG[5]. We also extract the Landé $g$-factor as $g = 2.001 \pm 0.001$ and the effective magnetization $\mu_0 M_{eff} = −445.2 \pm 0.4$ mT. The value of $g$ is close to the free electron value of 2.0, which implies low spin-orbit coupling. This is not surprising as magnetism in LAFO arises primarily from $Fe^{3+}$ (with $L = 0$). The value of $M_{eff}$ is also in excellent agreement with the field required to saturate along [100] as seen in Fig. 3a. In the past, spinel structure magnetic insulators were thought to be advantageous over the garnets due to their simpler crystal structure and lower synthesis temperature, but their damping values were consistently higher[22,28,29,31]. Our results show that spinel FMIs can also possess damping values that are competitive with the garnets.

## Spin-orbit torque switching

To demonstrate SOT switching in LAFO, we interface our magnet with a high spin-orbit coupled metal, Pt. The Pt layer was deposited via

room temperature sputtering on top of LAFO films. The critical current density required for current-induced SOT switching depends on a number of factors, including the anisotropy strength and the spin-charge conversion efficiency of the Pt/LAFO interface. In order to spin-orbit torque switch the LAFO film, we found that minimization of the LAFO thickness, accompanied by a weaker perpendicular magnetic anisotropy, reduces the critical current density for spin-orbit torque switching. Therefore we focus on the 4.1 nm samples in this section. Pt(2 nm)/LAFO(4.1 nm) Hall bars exhibit critical current densities as low as $6 \times 10^5$ A/cm², one of the lowest observed to date for PMA FMIs at room temperature. Figure 4a shows the Hall resistance, $R_{xy}$, measured as a function of an out-of-plane magnetic field, $H_z$, with the linear ordinary Hall contribution subtracted out. The presence of a hysteretic anomalous Hall effect is observed and likely emerges from the transfer of spin angular momentum across the Pt/LAFO interface and allows us to distinguish the up and down magnetization states[32]. A charge current passing through the Pt generates a spin current that travels towards the LAFO via the spin Hall effect. The direction of the moments in the LAFO imposes the boundary condition at the interface for spin accumulation, which in turn modifies the spin current in the Pt. The modified spin current in the Pt then produces additional transverse voltages via the inverse spin Hall effect, which is detected as a "spin Hall" anomalous Hall effect. Supplementary Material Section 4 provides a more detailed discussion of spin Hall magnetoresistance effects in Pt/LAFO bilayers.

We apply 5 ms-long DC current pulses along the Hall bar and measure $R_{xy}$ after each pulse in the presence of a small in-plane field, $H_x$, oriented along the current injection axis. Figure 4b shows $R_{xy}$ as a function of the pulsed current density, $J_{DC}$, in a field of $H_x = \pm 3$ mT with a critical current density $|J_c| \approx 1.5 \times 10^6$ A/cm². Typical critical current densities at this field for 4 nm LAFO films range from $1 - 1.5 \times 10^6$ A/cm². Also as seen in Fig. 4b, the switching polarity reverses direction on reversing the applied field direction $+ H_x \rightarrow − H_x$ as expected from the SOT switching mechanism.

To show that the switching is repeatable, we show $R_{xy}$ measured during a sequence of current pulses with alternating sign, and magnitude of $7.5 \times 10^6$ A/cm² in Fig. 4c. The value of $R_{xy}$ switches with the pulses and recovers nearly 100% of its full value of about $\pm 3.1$ mΩ after each pulse, demonstrating consistent and reversible switching. By performing the same switching experiments at different in-plane

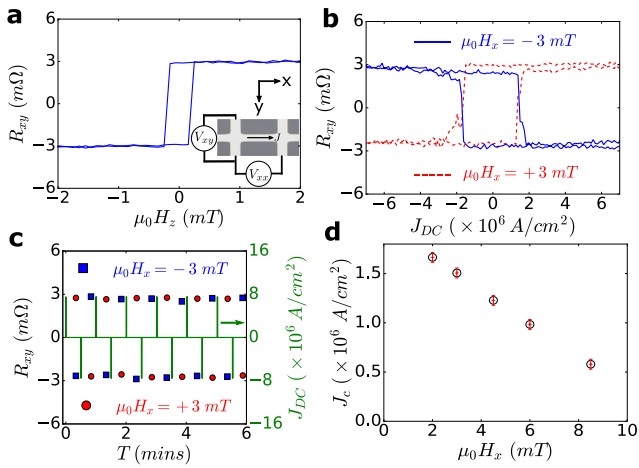

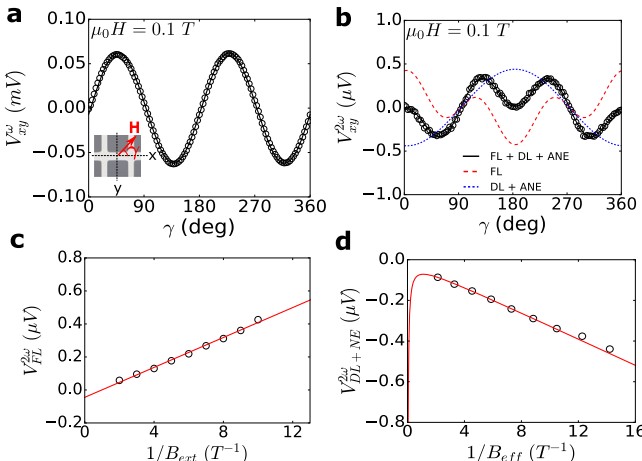

**Fig. 4 | Demonstration of current induced spin-orbit torque switching of the magnetization in a 4.1 nm LAFO film at room temperature. a** Anomalous Hall effect in the Pt showing clear hysteresis from the magnetization of the adjacent LAFO layer. Inset: optical image of the Hall bar with measurement geometry. The Hall bar dimensions are $10\,\mu m \times 40\,\mu m$. **b** $R_{xy}$ measured as a function of $J_{DC}$ at $H_x$ of $\pm 3$ mT along the current direction. **c** $R_{xy}$ measured with a sequence of current pulses of alternating sign for $H_x$ of $\pm 3$ mT along the current direction. **d** Critical switching current density $J_c$ as a function of in-plane field. The inset shows the switching loops for $\mu_0 H_x = 2.0$ mT and $\mu_0 H_x = 8.5$ mT. The uncertainty is obtained from the width of the transition.

**Fig. 5 | First and second harmonic Hall measurements for extraction of $B_{DL}$ and $B_{FL}$. a** First harmonic Hall voltage $V_{xy}^\omega$ as a function of in-plane angle $\gamma$ (inset, an optical image of the patterned Hall bar). The solid black line is a fit to Eq. (1). **b** Second harmonic Hall voltage $V_{xy}^{2\omega}$ as a function of $\gamma$. The solid black line is a fit to Eq. (2), and the solid red and blue lines denote the field-like and damping-like + Nernst effects (ordinary and anomalous) contributions respectively. **c, d** The field-like and damping-like + ANE contributions to $V_{xy}^{2\omega}$ as a function of $1/B_{ext}$ and $1/B_{eff}$ respectively as described in the main text. The red lines are linear fits. Measurements were performed with an AC current of 1 mA amplitude.

fields, we observe $J_c$ monotonically decrease with increasing $H_x$ as expected[33]. This is shown in Fig. 4d, where $J_c$ as low as $6 \times 10^5$ A/cm$^2$ is achieved for $H_x = 8.5$ mT. As a comparison, a study of a YIG(5 nm)/Pt system reported a switching current of $3 \times 10^7$ A/cm$^2$ for $H_x = 5.0$ mT, more than an order of magnitude larger than the corresponding value in our system (Fig. 4d)[23]. Other studies in YIG and thulium iron garnet systems have reported $J_c$ values on the order of $10^7$ A/cm$^2$ at fields on the order of a few tens of mT[5,24]. We note however that a comparison of $J_c$ across different systems is not meaningful as the value of $J_c$ also depends strongly on the strength of the PMA and the thicknesses of the magnetic and Pt layers. We also performed SOT switching in a 15 nm LAFO film (Supplementary Materials Section 7), where the $J_c$ is comparable to those reported in Pt/YIG systems due to the larger anisotropy of thicker LAFO films. However, LAFO provides a rare combination of weak PMA at ultra-low thicknesses, both of which help lower $J_c$. This combination allows the realization of ultra-low $J_c$ at low thicknesses.

The presence of an in-plane field is necessary to break the symmetry in order to achieve deterministic switching, but is detrimental for device applications. Field-free switching has been achieved in select systems by breaking the symmetry in other ways, including using exchange bias and physically engineering an asymmetric stack[34,35]. We hope that similar techniques can be incorporated with LAFO in the near future to increase its practicality.

The SOT switching can also be characterized by SOT efficiency which is a measure of spin to charge conversion of the Pt/LAFO bilayer. SOT efficiency takes into account the spin transparency of the interface and spin Hall angle of Pt and can be estimated with in-plane angular harmonic Hall measurements. An in-plane magnetic field $B_{ext} = \mu_0 H_{ext}$ larger than the anisotropy (see Supplementary Material Sections 4 and 5) is applied at an angle $\gamma$ with respect to the current channel (Fig. 5a inset) and an AC current is used to measure the Hall voltages. For a system with low in-plane anisotropy, the first ($V_{xy}^\omega$) and second ($V_{xy}^{2\omega}$) harmonic Hall voltages are given by[36–38]

$$V_{xy}^\omega = I_{rms} R_{PHE} \sin 2\gamma, \tag{1}$$

$$V_{xy}^{2\omega} = V_{FL}^{2\omega} \cos 2\gamma \cos \gamma + V_{DL\,+NE}^{2\omega} \cos \gamma,$$
$$V_{FL}^{2\omega} = -\left(\frac{B_{FL}}{B_{ext}}\right) R_{PHE} I_{rms}, \tag{2}$$
$$V_{DL\,+NE}^{2\omega} = -\left[\frac{B_{DL}}{2B_{eff}} R_{AHE} + \alpha_{ANE} + \beta_{ONE} B_{ext}\right] I_{rms},$$

where $I_{rms}$ is the RMS amplitude of the AC current, $R_{PHE}$ and $R_{AHE}$ are amplitudes of the planar Hall and anomalous Hall effect respectively, $B_{FL}$ and $B_{DL}$ are the field-like (FL) and damping-like (DL) effective fields associated with the spin-orbit torque, $\alpha_{ANE}$ is a term characterizing the parasitic contribution arising from the anomalous Nernst effect (ANE), $\beta_{ONE}$ is a term characterizing the contribution from the ordinary Nernst effect (ONE), and $B_{eff} = B_{ext} - \mu_0 M_{eff}$ where $M_{eff}$ is the effective magnetization. Figure 5a shows the angular dependence of $V_{xy}^\omega$ at $B_{ext} = 0.1$ T. We fit this to extract $R_{PHE} = 92 \pm 4$ m$\Omega$ as per Eq. (1). The relatively large uncertainty is due to a slight variation of $R_{PHE}$ for different applied fields. $R_{AHE} \approx 3.1$ m$\Omega$ was obtained by measuring $R_{xy}$ using an out-of-plane field as shown in Fig. 4a. To obtain $B_{FL}$ and $B_{DL}$, we fit the angular dependence of $V_{xy}^{2\omega}$, as shown in Fig. 5b, to extract the FL and DL + Nernst effect (ANE and ONE) contributions, $V_{FL}^{2\omega}$ and $V_{DL\,+NE}^{2\omega}$ as per Eq. (2). Figure 5c and d show the inverse field dependence of these contributions, whose fit allow us to extract $B_{FL}$ and $B_{DL}$ as $-0.70 \pm 0.03$ mT and $30.5 \pm 0.8$ mT per $J_{rms} \approx 3.5 \times 10^6$ A/cm$^2$ respectively. From the fit in Fig. 5d we also obtain $\alpha_{ANE} = 19 \pm 3$ nV and $\beta_{ONE} = -40 \pm 20$ nV/T, indicating that the Nernst contributions and current heating are negligible.

To quantify the efficiency of spin-to-charge conversion in our system, we calculate the DL and FL spin torque efficiencies $\theta_{DL,FL}$ as[37]

$$\theta_{DL(FL)} = \left(\frac{2|e|}{\hbar}\right)\left(\frac{M_s t_{LAFO} |B_{DL(FL)}|}{J_{Pt}}\right), \tag{3}$$

where $e$ is the electron charge, $\hbar$ is the reduced Planck's constant, $M_s$ is the saturation magnetization, $t_{LAFO}$ is the thickness of the LAFO layer, and $J_{Pt}$ is the current density amplitude in the Pt. From this, we obtain $\theta_{DL} = 0.57 \pm 0.01$ and $\theta_{FL} = 0.013 \pm 0.001$. The large value of $\theta_{DL}$ is consistent with the $J_c$ required for switching. We do however note that the second harmonic Hall technique has a tendency to overestimate the SOT efficiencies[39].

## Discussion

A recent study on a different Pt/FMI spinel system using $Mg(Al,Fe)_2O_4$ (MAFO) found an above average damping-like SOT efficiency of $\theta_{DL} \approx 0.15$[40]. However, the damping-like SOT efficiency in Pt/MAFO is still lower than that observed in Pt/LAFO, indicating that Pt epitaxy alone is insufficient to explain the exceptional charge-to-spin conversion efficiency in Pt/LAFO bilayers. One key difference between MAFO and LAFO is that a 2 nm thick magnetically dead layer at the MAFO film/substrate interface limits film quality in the ultra-thin regime[30]. Such magnetic defects can prevent the entirety of a film from being switched uniformly with magnetic field or current. In contrast, the minimal defects in LAFO/MGO makes it a much cleaner system for SOT switching. Bulk saturation magnetization values and the absence of dead layers at a few unit cells of LAFO allows the SOT to act on the entirety of a magnetically uniform, high quality ultra-thin LAFO film, and are contributing factors to the large $\theta_{DL}$. The high quality Pt/LAFO interface can also facilitate the efficient transfer of spin across it. For instance, it is known in YIG that a poor interface with the Pt results in poor spin transfer[41].

In summary, we have demonstrated a promising new class of nanometer-thick low damping spinel ferrite thin films with highly efficient current-induced SOT switching. These LAFO films on MGO exhibit critical switching current densities as low as $6 \times 10^5$ A/cm² when interfaced with Pt and a damping-like SOT efficiency as high as $\theta_{DL} \approx 0.57$. This superior performance was attributed to a combination of the excellent epitaxial quality of LAFO in the ultra-thin regime, the epitaxial growth of the Pt overlayer, and the high quality Pt/LAFO interface. LAFO also has been shown to have one of the lowest magnetic damping values of PMA FMIs to date, making it promising in numerous other applications. Altogether LAFO on MGO is the first demonstration of all of the desirable properties of low damping PMA materials in one material, and represents an unprecedented step towards the realization of a new type of spin-wave material platform for the next generation of spintronic devices.

## Methods

### Sample fabrication

Films of LAFO were synthesized via pulsed laser deposition (PLD) with a KrF laser ($\lambda = 248$ nm) on (001)-oriented single crystal MGO substrates. The MGO substrates of size $5 \times 5 \times 0.5$ mm³ were prepared from high quality bulk single crystals grown by the Czochralski method at the Leibniz-Institut für Kristallzüchtung, Berlin, Germany, as described in detail elsewhere[42–44]. A pressed $Li_{0.6}Al_{1.0}Fe_{1.5}O_4$ target was used for ablation, which includes an additional 0.1 Li enrichment to compensate for Li loss during deposition due to Li volatility. Prior to deposition, the target was pre-ablated at 1 Hz for 1 min, followed by 5 Hz for 1.5 min in vacuum along a circular track of radius 0.75 cm at a laser fluence of ~1.9 J/cm². The substrates were cleaned via sonication in acetone and isopropanol for 10 min each. The deposition was then performed on the pre-ablated track in a 15 mTorr $O_2$ atmosphere with a substrate temperature of 450 °C, target-to-substrate distance of 3 in, and a laser fluence of 2.8 J/cm² operating at 2 Hz. The laser spot size was 6 mm². After deposition, the substrate was left to cool to ambient temperature in 100 Torr $O_2$. These deposition parameters give rise to a growth rate of ~0.0125 nm/pulse, and the resulting films are insulating with resistances greater than our measurement limit of 1 GΩ. It is worth noting that the synthesis temperature of LAFO is considerably lower than the 600–900 °C required for low damping epitaxial garnet films[45–47]. This makes LAFO compatible with a wider range of materials that can not tolerate high processing temperatures.

### HAADF-STEM imaging

The HAADF-STEM imaging was performed using a Titan 60-300 TEM operated at an accelerating voltage of 300 kV. Samples for cross-sectional transmission electron microscopy were prepared by focused ion-beam (FIB) milling using Ga-ion source. Prior to TEM observations an additional Ar-ion polishing at low voltage (500–700 V) was performed in order to remove the residual surface Ga and reduce FIB-induced sample roughness.

### FMR measurements

Broadband ferromagnetic resonance (FMR) measurements were performed on a custom built FMR setup consisting of a copper waveguide with a center conductor width of 250 μm between two electromagnets. A small modulation field of 2–3 Oe was applied on top of the DC field and a lock-in amplifier was used for signal detection after filtering through a microwave diode. The measured signal is the absorption derivative $dI_{fmr}/dH$, which is then fit to a Lorentzian derivative of the form

$$\frac{dI_{fmr}}{dH} = -A \left( \frac{2(H - H_{fmr})\Delta H_{hwhm}}{\left( \Delta H_{hwhm}^2 + (H - H_{fmr})^2 \right)^2} \right) \tag{4}$$

$$+ D \left( \frac{(H - H_{fmr})^2 - \Delta H_{hwhm}^2}{\left( \Delta H_{hwhm}^2 + (H - H_{fmr})^2 \right)^2} \right), \tag{5}$$

where the first and second terms represent the absorptive and dispersive components of the FMR spectrum respectively. From this fit, we can extract the FMR half-width-half-maximum linewidth $\Delta H_{hwhm}$ and the FMR resonance field $H_{fmr}$.

### Second harmonic measurements

Second harmonic Hall measurements were performed with an AC current of frequency $\omega/2\pi = 524.1$ Hz and RMS density $J_{rms} \approx 3.5 \times 10^6$ A/cm². The first ($V_{xy}^{\omega}$) and second ($V_{xy}^{2\omega}$) harmonic Hall voltages are then simultaneously measured using lock-in amplifiers as a function of in-plane angle $\gamma$. Note that $V_{xy}^{\omega}$ is the in-phase component whereas $V_{xy}^{2\omega}$ is the quadrature component.

### XAS and XMCD

Room temperature XAS and XMCD were performed at beamline 4.0.2 of the Advanced Light Source, Lawrence Berkeley National Laboratory. A magnetic field of ±0.1 T was applied perpendicular to the film plane and the X-ray absorption spectra was measured as a function of energy with the X-ray beam fixed at negative circular polarization.

## Data availability

The datasets generated during and/or analysed during the current study are available from the corresponding author on reasonable request.

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

## Acknowledgements

We thank Satoru Emori for helpful discussions and Jutta Schwarzkopf for a critical reading of the paper. This work was supported by the U.S. Department of Energy, Director, Office of Science, Office of Basic Energy Sciences, Division of Materials Sciences and Engineering under Contract No. DESC0008505. S.C. and L.J.R. were supported by the Air Force Office of Scientific Research under Grant No. FA 9550-20-1-0293. J.J.W. was supported by the National Science Foundation under Award DMR-2037652. L.J.R. was also supported by an NSF Graduate Fellowship. H.R. was supported by Quantum Materials for Energy Efficient Neuromorphic Computing (Q-MEEN-C), an Energy Frontier Research Center funded by the US Department of Energy (DOE), Office of Science,

Basic Energy Sciences (BES), under Award DE-SC0019273. E.C. and A.D.K. were supported by the National Science Foundation under Award DMR-2105114. This research used resources of the Advanced Light Source, a U.S. DOE Office of Science User Facility under contract no. DE-AC02-05CH11231. X-ray diffraction was performed at the Stanford Nano Shared Facilities at Stanford University, supported by the National Science Foundation under Award No. ECCS-1542152. Krishnamurthy Mahalingam and Cynthia Bowers acknowledge funding support from the Air Force Research Laboratory under Awards: AFRL/NEMO: FA8650-19-F-5403 TO3 and AFRL/MCF: FA8650-18-C-5291, respectively. Research at NYU was supported by NSF DMR-2105114.

## Author contributions

X.Y.Z., S.C. and Y.S. conceived of the research ideas; X.Y.Z., S.C., L.J.R. and J.J.W. and A.V. conducted X-ray and static magnetic characterization, AFM, FMR and transport measurements; K.M., C.T.B. and M.E.M. conducted TEM measurements; A.T.N. conducted XMCD measurements; E.C., H.R., and A.D.K sputtered Pt on the films; Z.G. supplied the MGO substrates; X.Y.Z., S.C. and Y.S. wrote the manuscript; Y.S. supervised the research.

## Competing interests

The authors declare no competing interests.
