## [Peer Review File · Nature Communications]

Reviewers' Comments:

Reviewer #1:

Remarks to the Author:

In their manuscript the authors describe thin films of Lithium Aluminate Spinel ferrite films with interesting magnetic properties. Among these properties are low Gilbert damping, perpendicular magnetic anisotropy (PMA) and easy switching by spin-orbit torque. The claims are supported by X-ray diffractometry and reflectivity, transmission electron microscopy, SQUID magnetometry, ferromagnetic resonance studies, Hall effect measurements and harmonic Hall measurements. The studies are well carried out and give a detailed insight in the magnetic and structural quality of the samples.

The work is significant for the field. The results are impressive and show that there are materials besides the garnets that can have lower damping than ferromagnetic metals. Also the SOT switching is a useful property. Nevertheless, the significance is not as high as the authors indicate. Although the damping is quite low, it cannot compete with the results from garnets published in literature despite the authors' claims.

Here is also the main point of criticism because a major claim is that the results indeed compare to those known from garnets or are even superior. These claims are unjustified and it would probably show the results in a much better light if the really good numbers were put in the right context (not as good as garnets but better than ferromagnetic metals). Also in some places, these statements are not justified by the proper references.

This concerns the following claims:

On page 1 right hand side the authors claim that Garnet thin films usually exhibit a dead layer. No reference is given for this and for example Chang et al. IEEE MAGNETICS LETTERS, Volume 5 (2014) show a 22 nm thin film with excellent properties and no sign of a dead layer. Also d'Allivy Kelly (Ref. 22) et al. report identical magnetizations for 20 nm and 7 nm thin films indicating no dead layer.

On page 2 it is claimed that the damping of $6e-4$ is among the lowest for layers with PMA. It is omitted here that this damping does not occur for the thinnest films but only for a thickness of 15 nm which is not necessarily ultrathin (see below). In addition Ref. 14 shows a damping that is a factor of 2 smaller ($3e-4$) which is not negligible. In Ref. 5 Ding et al. also report a damping of $4e-4$ for an 8 nm thin film with perpendicular anisotropy. The authors on the other hand achieve even a higher damping only for the 15 nm thin film.

On page 3 top right paragraph the authors even claim that the damping is similar to "those reported in the garnets". This is really not the case. The best damping reported to date for thin film YIG is from Ding et al (IEEE MAGNETICS LETTERS, Volume 11 (2020)) who demonstrate a Gilbert damping of $5e-5$ which is more than an order of magnitude lower than that demonstrated here. And also other examples in the list of references show a damping better than $1e-4$.

Even for ultrathin films Dubs et al. (PHYSICAL REVIEW MATERIALS 4, 024416 (2020)) show a damping of $1.2e-4$ (11 nm thin film).

Also the inhomogeneous broadening for all the films presented here (and especially the thin ones) is more than 1 mT while YIG with in-plane anisotropy can achieve values of less than 0.1 mT (Dubs et al.).

So I suggest that the authors correct their claim by saying that though the damping and inhomogeneous broadening does not compare to that of thin film YIG, it is almost as good as that reported for YIG with perpendicular anisotropy and definitely better than for ferromagnetic metals which is impressive enough without overstating.

Besides this general remark I have a few smaller ones:

The hysteresis loops of the SQUID measurements measured in-plane have an opening around zero field. This should be discussed in the text.

The presence of the hysteretic anomalous Hall-effect is justified by transfer of spin angular momentum at the LAFO/Pt interface. This is correct. The reference to the SHMR also is, but it would be useful to indicate that in the reference this is discussed as SHAHE and a few words about the origin would also help because the manuscript should address a broader audience.

The SOT switching also looks convincing. In order to assess it, the numbers for the current density should be put in context with other results and appropriate citations. The usefulness should be briefly discussed if for the lowest current density the highest in-plane field is needed. Is a switching without external field possible? Is this useful for applications? Can a ferromagnetic insulator be used for example for reasonable memory devices?

The claim that the semi-epitaxial quality of the Pt is a reason for the good spin transmission at the interface also sounds convincing. However, it is also known from garnets that a non-stoichiometry at the surface can reduce the spin transmission. So just a better interface quality may be a reason just as well. At least alternatives should be discussed.

I do not think that the manuscript should be published in the present format. Especially the claims should be put in the right context and a sound comparison to garnets should be done. Simply comparing the numbers rather than general statements like "among the best values" would give a clear picture that can be judged by the reader.

Besides this, I think that the work is carefully done. The manuscript would even be acceptable with less methods used and the description of the methods is detailed enough to reproduce the results. Only for the SQUID measurements, a few more details might be good. The measurement of the volume magnetization is an important claim. To me the error bars seem at the lower limit of what can be achieved. It might be good to mention, what error can be assumed for the thickness and what error is assumed for the SQUID measurements (it must be pretty small) and how this was achieved. SQUID magnetometry is very sensitive but also prone to errors that can easily reach 10 or 20%, unless a careful calibration with for example a properly sized reference sample is performed.

Reviewer #2:

Remarks to the Author:

The manuscript entitled "Ultra-Thin Lithium Aluminate Spinel Ferrite Films with Perpendicular Magnetic Anisotropy and Low Damping" from X.Y. Zhang et al. , describes the growth, magnetic characterization and magnetization switching by spin-orbit torque in nanometer thick films of spinel structures $\text{Li}_{0.5}\text{Al}_{1.0}\text{Fe}_{1.5}\text{O}_4$ (LAFO) on MgGa_2O_4 (MGO) substrates. The authors report on unprecedented low magnetic damping for these films and thicknesses (in the order of $\alpha_{\text{Gilbert}}=6 \times 10^{-4}$) in the presence of perpendicular magnetic anisotropy for the pure films. The study on this material system is enriched by the addition of a heavy metal layer (Pt) on the LAFO layer and (current induced) spin-orbit torque induced magnetization switching via first and second harmonic Hall measurements and large damping-like spin torque efficiencies are reported. The supplementary material gives further background, more detailed information on the different aspects of the experiment.

While spinel ferrites are known since decades, the synthesis of new types of spine ferrite films remains a challenge and, even more importantly the development of (spintronic) applications based on spinel ferrite thin films appears to remain an unsolved question so far. The rich number of different properties of spinel ferrite based thin films renders them to have potential for various areas, from information technology such as magnetic sensing and data storage to magnetization switching and spintronic devices and their electrical application for random access memories and energy storage devices. However, higher values of magnetic damping and comparably high critical currents for magnetization switching via spin-orbit torque impede the efficient realization of above applications.

The work of the authors in the present manuscript does indeed fill a gap, by introducing a new class of ultra-thin film LAFO/MGO films exhibiting also PMA which is particularly interesting both for spintronics and magnonics and which has not been done before to the best of my knowledge. I just need to note, that there is another work published in Nat.Comm. from a similar group of authors reporting on hybrid STNOs using LAFO films as well, however the magnetic damping reported there seems to be higher than in the current manuscript (H. Ren et al., Nat. Commun. 14, 1406 (2023)) and it is not used as a standalone material class. In absolute units, the critical switching currents in films with PMA are in a similar magnitude as for a similar work on a single layer in GaMnAs (M.Jiang et al., Nat. Commun. 10, 2590 (2019)) but the authors system is a ferromagnetic insulator, not a semiconductor.

In general, the paper is really well and clearly written, and also easily understandable for the non-expert reader who is not directly involved in research of spinel ferrites. In my opinion the presented data is – convincing, presented in a logically, sound manner and the manuscript timely as other low damping insulating ferromagnets, especially with PMA than YIG, for instance, are needed. Thus, it presents " important advances of significance" in the field of spintronics and magnonics.

Therefore, in view of all above points, in principle I think that this manuscript is suited for

publication in Nature Communication after minor revision.

On main critical point to me is that in my opinion the authors are not fully consistent with the discussion of their results regarding the sample thickness. By looking into the supplemental information, the authors utilized four different thicknesses. For instance, they show the FMR for the 15.1 nm thick film which is also nicely in the material characterization in Figure 1 but for the current switching they discuss the results from a film thickness of 4.1 nm, which was of lower quality at least in the 2θ - ω scan. Could the authors please comment on that, either by changing the figures in the main to one choice of thickness if there is a thickness dependence, if there is not by showing that absence?

I have some remaining question/remarks s I would like to kindly ask to the authors to address:

1. Generally the texts in the Figures, especially in the insets or text descriptions, the fontsize is by far too small and almost not readable in a print-out.
2. In black-white printing a lot of different curves are not distinguishable as they appear to be the same, maybe change the line style font as well.
3. In some figures and text the contrast seems to be not really good and a bit blurry.
4. In Figure 1 (a) the authors write MGO (110) but in Figure 1 (c) MGO 115. As a non-expert for these type of images, could the authors please clarify a bit here as this seems confusing to me. Also in the caption the authors write the image is along [110] but put round brackets in (a).
5. Is the sample fully strained out-of-plane as well, is there still some remaining compressive strain, leading to a shrinking of the height in z-direction?
6. The exact origin of the PMA in their system does not become clear in the main, but is clearly discussed in the supplemental. To me this is an important information and maybe the authors could also add a corresponding remark into the main.
7. On page 2 the authors write "critical switching currents as low as ...", I would specify this is for an ferromagnetic insulator (see paper reference above) and also give other typical values from literature to give a better understanding of the exact number to a broad community.
8. Can the authors show the $w(H)$ dependence in the main as well?
9. In part B, and generally in the paper, the variety of used film thicknesses for this study does not become clear, can the authors add these please to the discussion in number than merely referring to the supplemental.
10. Did the authors consider other contributions in their system in addition to the SOT switching/other spin-to-charge? Heating by the current application, does it lead to temperature gradients? In the supplemental other magnetoresistive effects are discussed can the authors add a reference to it in the main, please?
11. What is the spin-mixing conductance at the LAFO/Pt interface?
12. In the supplementary material, part 1 the authors write "smooth interface", what means smooth, could they please specify in terms of surface roughness, for instance?

In general, similar comments regarding figures etc. also apply to the supplementary material.

Yours sincerely,
Reviewer

Xin Yu Zheng
Geballe Laboratory for Advanced Materials
McCullough Bldg. Rm 175
476 Lomita Mall
Stanford University
Stanford, CA 94305-4045
TEL: (650) 724-4007
xinzheng@stanford.edu

We thank both referees for critically reading our manuscript and providing thoughtful feedback that has allowed us to improve our manuscript. We appreciate that Referee A notes that our work is “*significant for the field*” and Referee B states that our work “*fills a gap, by introducing a new class of ultra-thin films ... exhibiting also PMA which is particularly interesting both for spintronics and magnonics.*”

We address each of the concerns of both referees below and describe the associated changes to the manuscript. With these changes, we believe that our manuscript is now ready for publication in *Nature Communications*.

REFEREE A

1. Referee A states that “*The work is significant for the field. The results are impressive and show that there are materials besides the garnets that can have lower damping than ferromagnetic metals. Also the SOT switching is a useful property. Nevertheless, the significance is not as high as the authors indicate. Although the damping is quite low, it cannot compete with the results from garnets published in literature despite the authors' claims. Here is also the main point of criticism because a major claim is that the results indeed compare to those known from garnets or are even superior. These claims are unjustified and it would probably show the results in a much better light if the really good numbers were put in the right context (not as good as garnets but better than ferromagnetic metals). Also in some places, these statements are not justified by the proper references.*”

We thank the referee for recognizing the significance of our work. We appreciate their pointing out inaccuracies in benchmarking our results. It is not our intention to overstate results but rather to introduce LAFO as a novel FMI material with ideal properties for studying spintronics and magnonics. Research in this field has been largely dominated by the garnets and so the presence of new materials can provide new ways of probing and understanding spin physics.

We have modified the manuscript by rephrasing statements and adding numbers with appropriate citations on page 1 right column paragraph 2, page 2 left column paragraph 1, and page 3 right column paragraph 1. These edits are detailed below in points 2, 3, and 4.

2. “*On page 1 right hand side the authors claim that Garnet thin films usually exhibit a dead layer. No reference is given for this and for example Chang et al. IEEE MAGNETICS LETTERS, Volume 5 (2014) show a 22 nm thin film with excellent properties and no sign of a dead layer. Also d'Allivy Kelly (Ref. 22) et al. report identical magnetizations for 20 nm and 7 nm thin films indicating no dead layer.*”

We thank the referee for pointing out this oversight. A literature search reveals that claims about the presence of dead layers in garnet based systems are inconsistent. There are several studies that report the presence of dead layers or interdiffusion layers at the YIG/GGG interface [Sci. Rep. **7**, 11774 (2017); Phys. Rev. B **96**, 104404 (2017); Appl. Phys. Lett. **115**, 182401 (2019); Phys. Rev. Mater. **2**, 104404 (2018)]. Among these studies, the interdiffusion layer exhibits an absence or suppression of magnetization and has a thickness that ranges from 1.2 nm to 5-7 nm. We now cite these studies. However, we agree with the referee that other studies (including the two mentioned by the referee) report saturation magnetization values similar and, in some cases, greater than bulk values from which the absence of a dead or interdiffused layer at the film/substrate interface is deduced. Moreover we have found that there is one study that shows direct TEM imaging of a sharp interface in a TmIG/GGG system [Sci. Rep. **8**, 11087 (2018)].

In light of these conflicting results, we have modified our wording on page 1 right column second paragraph to become “*Although garnets are well known for their low damping, many studies have shown the presence of a significant magnetic dead layer at the film-substrate interface due to interdiffusion, likely a result of their high growth temperatures of 600-900 C. However some studies have shown that a sharp interface with bulk magnetization properties can be obtained in garnet films.*” We also have included citations that justifies the above rewording, showing studies where a dead layer is present.

3. “*On page 2 it is claimed that the damping of $6e-4$ is among the lowest for layers with PMA. It is omitted here that this damping does not occur for the thinnest films but only for a thickness of 15 nm which is not necessarily ultrathin (see below). In addition Ref. 14 shows a damping that is a factor of 2 smaller ($3e-4$) which is not negligible. In Ref. 5 Ding et al. also report a damping of $4e-4$ for an 8 nm thin film with perpendicular anisotropy. The authors on the other hand achieve even a higher damping only for the 15 nm thin film.*”

We have changed the wording on page 2 left column paragraph 1 to “*which is as low as 6×10^{-4} for 15 nm thick films and on the order of values reported in yttrium iron garnet systems with PMA at room temperature*” and cited Ref. 14 and Ref. 5 after this sentence.

4. “*On page 3 top right paragraph the authors even claim that the damping is similar to “those reported in the garnets”. This is really not the case. The best damping reported to date for thin film YIG is from Ding et al (IEEE MAGNETICS LETTERS, Volume 11 (2020)) who demonstrate a Gilbert damping of $5e-5$ which is more than an order of magnitude lower than that demonstrated here. And also other examples in the list of references show a damping better than $1e-4$. Even for ultrathin films Dubs et al. (PHYSICAL REVIEW MATERIALS 4, 024416 (2020)) show a damping of $1.2e-4$ (11 nm*

thin film).”

We would like to point out that the results by Ding et al. and Dubs et al. cited by the referee here are for easy-plane YIG films and not PMA ones. For films with PMA, the damping is slightly higher around $3-4 \times 10^{-4}$ (as pointed out by the referee in point 3 above) and the inhomogeneous broadening is similar to LAFO (Ding et al. in Ref. 5 report an inhomogeneous broadening of 3 mT in 8 nm YIG with PMA).

We have therefore changed the wording on page 3 right column paragraph 1 to “...and is approaching those reported in YIG with PMA” and cited some works on low damping PMA YIG after this sentence.

5. *“Also the inhomogeneous broadening for all the films presented here (and especially the thin ones) is more than 1 mT while YIG with in-plane anisotropy can achieve values of less than 0.1 mT (Dubs et al.). So I suggest that the authors correct their claim by saying that though the damping and inhomogeneous broadening does not compare to that of thin film YIG, it is almost as good as that reported for YIG with perpendicular anisotropy and definitely better than for ferromagnetic metals which is impressive enough without overstating.”*

We thank the referee for pointing out the differences in the inhomogeneous broadening values of PMA LAFO and in-plane garnets. Although the inhomogeneous broadening in LAFO is an order of magnitude larger than those in in-plane garnets, it is as good as those reported in PMA YIG. For instance, Ding et al. (Ref. 5) reports an inhomogeneous broadening in 8 nm YIG films of 3 mT. We have now included in page 3 right column last paragraph a sentence stating in the inhomogeneous broadening of 15.1 nm LAFO: “...and an inhomogeneous broadening of $\mu_0 \Delta H_0 = 1.5 \pm 0.1 \text{ mT}$.” We also included a sentence comparing this value to PMA YIG and cited Ref. 5: “*The inhomogeneous broadening is also similar to those observed in PMA YIG.*”

6. *“The hysteresis loops of the SQUID measurements measured in-plane have an opening around zero field. This should be discussed in the text.”*

We thank the referee for this suggestion. The small opening around zero field is due to a small misalignment of the sample with respect to the in-plane field. At zero field the moments are perpendicular to the film plane, but the small misalignment results in a small projection along the in-plane direction which is detected as a non-zero moment in-plane. The coercivity of the small opening is about 18 mT, with a remanence of about 6 kA/m. These values correspond to an approximate 5 degree misalignment.

On page 3 left column last paragraph, we have now added a sentence in the magnetic characterization section explaining the small opening: *“The small opening around zero field of the in-plane trace is due to a small misalignment of the sample with respect to the in-plane magnetic field.”*

7. *“The presence of the hysteretic anomalous Hall-effect is justified by transfer of spin angular momentum at the LAFO/Pt interface. This is correct. The reference to the SHMR also is, but it would be useful to indicate that in the reference this is discussed as SHAHE and a few words about the origin would also help because the manuscript should address a broader audience.”*

As suggested, we have now added a high level explanation of how the SHAHE arises at the bottom of the right column on page 3: *“A charge current passing through the Pt generates a spin current that travels towards the LAFO via the spin Hall effect. The direction of the moments in the LAFO imposes the boundary condition at the interface for spin accumulation, which in turn modifies the spin current in the Pt. The modified spin current in the Pt then produces additional transverse voltages via the inverse spin Hall effect, which is detected as a “spin Hall” anomalous Hall effect.”*

8. *“The SOT switching also looks convincing. In order to assess it, the numbers for the current density should be put in context with other results and appropriate citations. The usefulness should be briefly discussed if for the lowest current density the highest in-plane field is needed. Is a switching without external field possible? Is this useful for applications? Can a ferromagnetic insulator be used for example for reasonable memory devices?”*

We thank the referee for this suggestion. In answer to the referee’s questions, switching without an external field is not possible in our system as we require a field to break the symmetry to achieve deterministic switching. However, other studies [*Nature Nanotechnology* **11**, 878 (2016); *ACS Appl. Mater. Interfaces* **11**, 30446 (2019)] have broken this symmetry in other ways without an external field, *including using exchange bias and physically engineering an asymmetric stack*. We hope that these techniques can be incorporated with LAFO in the near future. We think the introduction of a brand new material into the field of spintronics along with LAFO’s record low switching current densities make it an excellent candidate for spintronics applications. Furthermore, the tunable PMA strength in LAFO as a function of thickness also makes it promising for memory storage devices.

We have now added a more complete comparison between our results and those in the

literature for similar systems and also connected our results to the broader context of device applications.

We have added the following on page 4, right column 1st paragraph: *“As a comparison, a study of a YIG (5 nm)/Pt system reported a switching current of 3×10^7 A/cm² for $H_x = 5.0$ mT, more than an order of magnitude larger than the corresponding value in our system (Fig.4 (d)). Other studies in YIG and thulium iron garnet systems have reported J_c values on the order of 10^7 A/cm² at fields on the order of a few tens of mT. We note however that a comparison of J_c across different systems is not meaningful as the value of J_c also depends strongly on the strength of the PMA and the thicknesses of the magnetic and Pt layers. However, LAFO provides a rare combination of weak PMA at ultra-low thicknesses, both of which help lower J_c .*

The presence of an in-plane field is necessary to break the symmetry in order to achieve deterministic switching, but is detrimental for device applications. Field-free switching has been achieved in select systems by breaking the symmetry in other ways, including using exchange bias and physically engineering an asymmetric stack. We hope that similar techniques can be incorporated with LAFO in the near future to increase its practicality.”

9. *“The claim that the semi-epitaxial quality of the Pt is a reason for the good spin transmission at the interface also sounds convincing. However, it is also known from garnets that a non-stoichiometry at the surface can reduce the spin transmission. So just a better interface quality may be a reason just as well. At least alternatives should be discussed.”*

We thank the referee for this suggestion. Indeed, a good interface is another reason for large spin transmission in our system, as it has been shown in YIG that degraded interface quality decreases the spin mixing conductance [*Applied Physics Letters* **103**, 092404 (2013)]. In the case of LAFO, the TEM of the Pt/LAFO interface shown in Figure 2 (a) demonstrates a superior interface where the transition occurs within one atomic layer, which is much better than typical heavy metal/oxide interfaces. The high quality of this interface is an alternative reason for the high spin transmission.

We have now expanded the reasons for good spin transmission in page 6 / column 1/ paragraphs 1 and 2 to include the possibility of a high quality Pt/LAFO interface and cited a work on YIG/Pt showing that a poor interface can reduce the spin transmission.

10. *“Only for the SQUID measurements, a few more details might be good.”*

We thank the referee for this suggestion. We have now added more details in the Magnetic Characterization section, including the instrument used, how and at what temperature the measurement was conducted, and the value of the saturation

magnetization. We have also included an explanation of the origin of the small opening near zero field on the in-plane measurement (point 6 above).

On page 3 left column bottom paragraph, we now state *“We performed SQUID magnetometry on LAFO films using an Evercool MPMS by Quantum Design at room temperature. We measure the magnetization as a function of field along the in-plane and out-of-plane directions. These measurements show that LAFO films exhibit PMA with bulk saturation magnetization values $M_s \approx 75$ kA/m at room temperature. ... The small opening around zero field of the in-plane trace is due to a small misalignment of the sample with respect to the in-plane magnetic field. The origin of the PMA is due to magnetoelastic coupling as a result of the epitaxial bi-axial tensile strain on the film imposed by the substrate (see Supplemental Material Section 3).”*

11. *“The measurement of the volume magnetization is an important claim. To me the error bars seem at the lower limit of what can be achieved. It might be good to mention, what error can be assumed for the thickness and what error is assumed for the SQUID measurements (it must be pretty small) and how this was achieved. SQUID magnetometry is very sensitive but also prone to errors that can easily reach 10 or 20%, unless a careful calibration with for example a properly sized reference sample is performed.”*

We have now included in supplemental section 2 an explanation of how the uncertainty in the magnetization was calculated in supplemental figure S3 (e):

“The error bars on M_s were estimated from the linear fits of the diamagnetic background from the substrate that were subtracted from the signal in the SQUID measurements (the intercept term). This error bar is typically on the order of 1%. The film area was determined by overlaying a transparent grid with 1 mm spacings on top of the sample and calculating the area of a (possibly irregular) polygon constructed from the sample borders. We estimate the errors of this method to be about 10%. Finally, the thickness was determined by fitting the XRD and XRR spectra of the films that has an uncertainty of less than 0.1 nm. The total uncertainty of the net magnetization is calculated by multiplying these quantities and propagating the uncertainties for each and is on the order of 10%”

REFEREE B

We first like to address the referee’s note regarding the other work on hybrid STNOs using LAFO films [H. Ren et al., Nat. Commun. **14**, 1406 (2023)]. This group is a close collaborator of ours. After our group had realized low damping LAFO films with PMA, we began collaborating

with several other groups in parallel in integrating LAFO into various devices/applications, while we proceeded to conduct SOT switching experiments on our end. The work by Ren et al. was the product of one of these collaborations which was published before our manuscript.

We have added a sentence on page 2, left column bottom paragraph acknowledging the work by Ren et al.: *“To this end, LAFO has already been incorporated in building hybrid spin-Hall nano-oscillators which are essential in accelerating spintronic applications.”*

We now address the referee’s specific concerns below.

1. *“One main critical point to me is that in my opinion the authors are not fully consistent with the discussion of their results regarding the sample thickness. By looking into the supplemental information, the authors utilized four different thicknesses. For instance, they show the FMR for the 15.1 nm thick film which is also nicely in the material characterization in Figure 1 but for the current switching they discuss the results from a film thickness of 4.1 nm, which was of lower quality at least in the 2θ - ω scan. Could the authors please comment on that, either by changing the figures in the main to one choice of thickness if there is a thickness dependence, if there is not by showing that absence?”*

We thank the referee for pointing out that we should be clearer on the different film thicknesses. One of the points of this manuscript is to present a new PMA ferromagnetic insulator thin film material. We present in sections A and B and associated supplemental sections, the structure and magnetic properties of thick and thin LAFO films. All of our LAFO films exhibit high crystalline quality and epitaxy. Even the broadened film peak for the 4.1 nm film in Figure 1(b) is not due to poor film quality, but rather due to the fact that the film is ultra-thin. Evidence of this is shown in the rocking curves of Supplemental figure S1, where the 4.1 nm film still exhibits a FWHM identical to that of the substrate (and to all other film thicknesses as well).

In Figure 3(a), we present field dependent magnetization data for the 15.1 nm sample to demonstrate the perpendicular magnetic anisotropy. The strength of this PMA varies with thickness, but saturation magnetization remains at bulk values for all thicknesses. In Figure 3(b), we show the corresponding dynamic properties of the 15.1 nm sample that exhibits the lowest magnetic damping values in the mid 10^{-4} . In the interest of space, we provide the static and dynamic magnetic properties for all thicknesses in the supplemental section and highlight the 15.1 nm film that has the lowest damping and large structural signal. For the purpose of current switching of magnetization, not only are thinner films easier to switch, the weaker PMA in the 4.1 nm samples facilitates the SOT switching experiment. Therefore SOT switching and second harmonic measurements were performed on the 4.1 nm samples. We have performed SOT switching on thicker films as well (data not shown), but the higher PMA and larger volume of the ferromagnet reduce the switching current density.

On page 2 right column paragraph 1, we have added a sentence explaining that the structural quality of the samples is excellent across the board and the broadening of the film peaks in thinner is not a reflection on the structural quality: *“Note that the broadening of the 4.1 nm film peak is not due to poor film quality, but rather due to the lower thickness.”*

On page 3 right column 1st paragraph, we explain that we observe bulk saturation magnetization values and perpendicular magnetic anisotropy in a range of film thicknesses but show only the 15.1 sample data in the manuscript with the other thicknesses in the supplement. We have also added a sentence at the beginning of the spin-orbit torque section to justify why we focus on the 4.1 nm sample: *“In order to spin-orbit torque switch the LAFO film, we found that minimization of the LAFO thickness, accompanied by a weaker perpendicular magnetic anisotropy, reduces the critical current density for spin-orbit torque switching. Therefore we focus on the 4.1nm samples in this section.”*

2. *“Generally the texts in the Figures, especially in the insets or text descriptions, the font size is by far too small and almost not readable in a print-out.”*

We have now made the text in the figures and insets larger in both the main manuscript and the supplementary materials.

3. *“In black-white printing a lot of different curves are not distinguishable as they appear to be the same, maybe change the line style font as well.”*

We have now also changed the line/symbol styles in all plots where more than one line/symbol is present in both the main manuscript and the supplementary materials.

4. *“In some figures and text the contrast seems to be not really good and a bit blurry.”*

We have now ensured that all figures are in encapsulated postscript format to enhance its resolution.

5. *“In Figure 1 (a) the authors write MGO (110) but in Figure 1 (c) MGO 115. As a non-expert for these type of images, could the authors please clarify a bit here as this seems confusing to me. Also in the caption the authors write the image is along [110] but put round brackets in (a).”*

The RSM in Figure 1 (c) was taken on an asymmetric peak with an out of plane component in order to capture information regarding both the in-plane and out-of-plane reciprocal vectors. The (110) peak does not have an out of plane component, so an RSM on (110) would not yield information about the out-of-plane lattice constant.

We have added a sentence clarifying this in the right column of page 2: *“Note that the RSM is taken on an asymmetric (115) peak with an out-of-plane component in order to capture information for both the in-plane and out-of-plane reciprocal vectors.”*

We have also corrected the label in Figure 1 (a) to square brackets.

6. *“Is the sample fully strained out-of-plane as well, is there still some remaining compressive strain, leading to a shrinking of the height in z-direction?”*

We note that the substrate imposes a bi-axial in-plane tensile strain on the film as the lattice constant of the substrate is larger than that of the film. The in-plane tensile strain leads to a tetragonal distortion out of plane (ie. a shrinkage along the z direction) while still preserving the volume. This shrinkage can be seen directly in Figure 1 (a), where the out of plane film peaks are located at a larger 2θ value compared to that of the bulk. Furthermore, from Figure S1 (e) of the supplement we can see no systematic change of this shrinkage as a function of film thickness, suggesting that the films are fully strained out of plane and are not relaxed up to 22.6 nm.

7. *“The exact origin of the PMA in their system does not become clear in the main, but is clearly discussed in the supplemental. To me this is an important information and maybe the authors could also add a corresponding remark into the main.”*

We have now added a sentence explaining the origin of the PMA and a reference to the supplement on page 3 top of right column: *“The origin of the PMA is due to magnetoelastic coupling as a result of the epitaxial bi-axial tensile strain on the film imposed by the substrate (see Supplemental Material Section 3).”*

8. *“On page 2 the authors write “critical switching currents as low as ...”, I would specify this is for a ferromagnetic insulator (see paper reference above) and also give other typical values from literature to give a better understanding of the exact number to a broad community.”*

We now added a clarification that this value is for a ferrimagnetic insulator system and provided typical values from literature for similar systems in the left column of page 2: *“Note that this value is for a FMI system and is an order of magnitude lower than typical values of 10^7 A/cm² typically observed in garnet/Pt systems at similar fields”*

9. *“Can the authors show the $w(H)$ dependence in the main as well?”*

We thank the reviewer for this suggestion. We have now included the frequency ($f = \omega/2\pi$) dependence of the resonance field in Figure 3 (d) of the main manuscript.

On page 3 right column last paragraph, we have also added an explanation of Figure 3 (c) and the parameters extracted from the fit: *“We also extract the Lande g-factor as $g = 2.001 \pm 0.001$ and the effective magnetization $\mu_0 M_{eff} = -445.2 \pm 0.4$ mT. The value of g is close to the free electron value of 2.0, which implies low spin-orbit coupling. This is not surprising as magnetism in LAFO arises primarily from Fe^{3+} (with $L=0$). The value of M_{eff} is also in excellent agreement with the field required to saturate along [100] as seen in Fig. 3 (a).”*

We have also added the analysis of the out-of-plane resonance field vs. frequency into section 3 of the supplement.

10. *“In part B, and generally in the paper, the variety of used film thicknesses for this study does not become clear, can the authors add these please to the discussion in number than merely referring to the supplemental.”*

We have addressed this concern which is similar to point 1 on page 2 right column at the bottom of first paragraph.

11. *“Did the authors consider other contributions in their system in addition to the SOT switching/other spin-to-charge? Heating by the current application, does it lead to temperature gradients? In the supplemental other magnetoresistive effects are discussed can the authors add a reference to it in the main, please?”*

We thank the referee for bringing up the important point of additional contributions that can spuriously mimic spin torque effects. We have taken into account effects of temperature gradients by including contributions of the ordinary Nernst effect (ONE) and anomalous Nernst effect (ANE) in the 2nd harmonic Hall analysis, as seen in Equation (2) and subsequent discussion in the manuscript. We extracted very small values of the respective parameters ($\alpha_{ANE} = 19$ nV, $\beta_{ONE} = -40$ nV/T), indicating that such effects are negligible.

In addition, we measured the magnetic anisotropy of LAFO films in the Pt/LAFO bilayers using low field spin Hall magnetoresistance measurements at various different current magnitudes. We saw no dependence in the extracted anisotropies and hence believe there are no significant current-induced heating effects. These measurements are presented and discussed in Supplemental Section 5 and Figure S7.

We have now added a sentence on page 5 right column paragraph 1 explaining that the effects of temperature gradients and current heating are negligible. Moreover we have referenced the supplemental section on magnetoresistance effects on page 4 left column paragraph 1: *“Supplemental Material Section 4 provides a greater discussion of spin Hall magnetoresistance effects in Pt/LAFO bilayers.”*

12. *“What is the spin-mixing conductance at the LAFO/Pt interface?”*

From the change in the value of the magnetic damping on adding the 2nm Pt overlayer and assuming a typical Pt spin diffusion length of 1 nm, we can estimate the spin-mixing conductance as $\sim 3 \times 10^{14} \Omega^{-1} m^{-2}$. It should be pointed out that a more accurate value of both the spin diffusion length and spin-mixing conductance must be extracted from a Pt thickness dependence of the magnetic damping.

We have now included the estimated spin mixing conductance in section 4 of the Supplement.

13. *“In the supplementary material, part 1 the authors write “smooth interface”, what means smooth, could they please specify in terms of surface roughness, for instance?”*

We have now added a sentence in part 1 of the supplement stating the RMS roughness of each interface: *“AFM performed on the MGO and LAFO surfaces have typical RMS roughness less than 0.1 nm.”*

14. *“In general, similar comments regarding figures etc. also apply to the supplementary material.”*

We have adopted the recommendations of referee B in points 2, 3, and 4 above to the supplementary material as well.

By addressing all the concerns of the reviewers and making the appropriate revisions to the manuscript, we believe that our manuscript is ready for publication in *Nature Communications*. We attach below a list of major modifications to the manuscript.

LIST OF MAJOR CHANGES:

1. **Page 1 right column:** *“Although garnets are well known for their low damping, some studies have shown the presence of a significant magnetic dead layer at the film-substrate interface due to interdiffusion, likely a result of their high growth temperatures of 600-900 C. However some studies have shown that a sharp interface with bulk magnetization properties can be obtained in garnet films.”*
2. **Page 2 left column:** *“...which is as low as 6×10^{-4} for 15 nm thick films and on the order of values reported in yttrium iron garnet systems with PMA at room temperature.”*
3. **Page 2 left column:** *“Note that this value is for a FMI system and is an order of magnitude lower than typical values of $10^7 A/cm^2$ typically observed in garnet/Pt systems at similar fields.”*

4. **Page 2 left column:** *“To this end, LAFO has already been incorporated in building hybrid spin-Hall nano-oscillators which are essential in accelerating spintronic applications.”*
5. **Page 2 right column:** *“Note that the broadening of the 4.1 nm film peak is not due to poor film quality, but rather due to the lower thickness.”*
6. **Page 2 right column:** *“Note that the RSM is taken on an asymmetric (115) peak with an out-of-plane component in order to capture information for both the in-plane and out-of-plane reciprocal vectors.”*
7. **Page 3 right column:** *“Note that the broadening of the 4.1 nm film peak is not due to poor film quality, but rather due to the lower thickness.”*
8. **Page 3 right column 1st paragraph:** *“...and is approaching those reported in YIG with PMA.”*
9. **Page 3 right column 1st paragraph:** *“We performed SQUID magnetometry on LAFO films using an Evercool MPMS by Quantum Design at room temperature. We measure the magnetization as a function of field along the in-plane and out-of-plane directions. These measurements show that LAFO films exhibit PMA with bulk saturation magnetization values $M_s \approx 75$ kA/m at room temperature. ... The small opening around zero field of the in-plane trace is due to a small misalignment of the sample with respect to the in-plane magnetic field. The origin of the PMA is due to magnetoelastic coupling as a result of the epitaxial bi-axial tensile strain on the film imposed by the substrate (see Supplemental Material Section 3).”*
10. **Page 3 right column last paragraph:** *“We also extract the Lande g-factor as $g = 2.001 \pm 0.001$ and the effective magnetization $\mu_0 M_{eff} = -445.2 \pm 0.4$ mT. The value of g is close to the free electron value of 2.0, which implies low spin-orbit coupling. This is not surprising as magnetism in LAFO arises primarily from Fe^{3+} (with $L=0$). The value of M_{eff} is also in excellent agreement with the field required to saturate along [100] as seen in Fig. 3 (a).”*
11. **Page 3 right column last paragraph:** *“...and an inhomogeneous broadening of $\mu_0 \Delta H_0 = 1.5 \pm 0.1$ mT.” “The inhomogeneous broadening is also similar to those observed in PMA YIG.”*
12. **Page 4 left column first paragraph:** *“A charge current passing through the Pt generates a spin current that travels towards the LAFO via the spin Hall effect. The direction of the moments in the LAFO imposes the boundary condition at the interface for spin accumulation, which in turn modifies the spin current in the Pt. The modified spin current in the Pt then produces additional transverse voltages via the inverse spin Hall effect, which is detected as a “spin Hall” anomalous Hall effect. Supplemental Material Section 4 provides a greater discussion of spin Hall magnetoresistance effects in Pt/LAFO bilayers.”*
13. **Page 4 right column bottom:** *“As a comparison, a study of a YIG(5 nm)/Pt system reported a switching current of 3×10^7 A/cm² for $H_x = 5.0$ mT, more than an order of*

magnitude larger than the corresponding value in our system (Fig.4 (d)). Other studies in YIG and thulium iron garnet systems have reported J_c values on the order of 10^7 A/cm² at fields on the order of a few tens of mT. We note however that a comparison of J_c across different systems is not meaningful as the value of J_c also depends strongly on the strength of the PMA and the thicknesses of the magnetic and Pt layers. However, LAFO provides a rare combination of weak PMA at ultra-low thicknesses, both of which help lower J_c .

The presence of an in-plane field is necessary to break the symmetry in order to achieve deterministic switching, but is detrimental for device applications. Field-free switching has been achieved in select systems by breaking the symmetry in other ways, including using exchange bias and physically engineering an asymmetric stack. We hope that similar techniques can be incorporated with LAFO in the near future to increase its practicality.

14. **Page 4 left column first paragraph:** *“In order to spin-orbit torque switch the LAFO film, we found that minimization of the LAFO thickness, accompanied by a weaker perpendicular magnetic anisotropy, reduces the critical current density for spin-orbit torque switching. Therefore we focus on the 4.1 nm samples in this section.”*
15. **Page 6 left column first paragraph:** *“The high quality Pt/LAFO interface can also facilitate the efficient transfer of spin across it. For instance, it is known in YIG that a poor interface with the Pt results in poor spin transfer.”*
16. **Page 6 left column summary paragraph:** *“This superior performance was attributed to a combination of the excellent epitaxial quality of LAFO in the ultra-thin regime, the epitaxial growth of the Pt overlayer, and the high quality Pt/LAFO interface.”*
17. **Supplemental Materials section 1:** *“AFM performed on the MGO and LAFO surfaces have typical RMS roughness less than 0.1 nm.”*
18. **Supplemental Materials section 2:** *“The error bars on M_s were estimated from the linear fits in subtracting off the diamagnetic background from the substrate in the SQUID measurements (the intercept term), which is typically on the order of 1%. The film area was determined by overlaying a transparent grid with 1 mm spacings on top of the sample and calculating the area of a (possibly irregular) polygon constructed from the sample borders. We estimate the errors of this method to be about 10%. Finally, the thickness was determined by fitting the XRD and XRR spectra of the films, and the uncertainty is typically less than 0.1 nm. The total uncertainty of the net magnetization is calculated by multiplying these quantities and propagating the uncertainties for each.”*
19. **Supplemental Materials section 4:** *“From the increase in damping with the Pt layer, we estimate that the spin mixing conductance at the interface is approximately $3 \times 10^{14} \Omega^{-1}m^{-2}$.”*
20. Texts in figures and insets have been made larger to make it easier to read. Figures that contain more than one line or symbol have changed line styles and symbol shapes so that they can be differentiated in a black and white printout.

21. Added the dependence of FMR resonance field on frequency as Figure 3 (d). Also added a close-up of the [001] SQUID trace near the origin as Figure 3 (b).
22. Removed the inset in Figure 4 (d), as it was too small to read and did not add any extra insight into the results.
23. Included a “data availability” section as per the requirements of *Nature Communications*.

Reviewers' Comments:

Reviewer #1:

Remarks to the Author:

The authors have addressed each of my concerns to my satisfaction. The paper is now suitable for publication without further revisions.

Reviewer #2:

Remarks to the Author:

I would first like to thank the authors for the revised version of the manuscript, where they included to a great extent my remarks.

However, there is still a minor point I would like to point out where I still disagree with the authors' response. I pointed out that the authors use different film thicknesses to show the different aspects of their study on this new material class. I understand that the best data is more favoured to be shown but, for instance, the answer to my critics on choosing the film for the current switching experiment is because

"for the purpose of current switching of magnetization, not only are thinner films easier to switch, the weaker PMA in the 4.1 nm samples facilitates the SOT switching experiment." as the authors write,

is not fully adequate in my opinion.

The authors stress the fact that indeed the critical switching current density is remarkably low and following the remark of the other reviewer compare it to a YIG/Pt system of 5nm thickness. First, since there is such strong thickness dependence- and they also say it themselves in the revised version - a true comparison is not possible, the compared values are already for a thicker film.

To compare more qualitatively: The authors mention they also measured the critical switching current density for different film thicknesses, do they also have data such as the one where they present the FMR study in the main? Would it be possible to show some dependency of the current density from the film thickness, for instance in the supplemental material? Then the difference in order of magnitude to the YIG/Pt might be also seen even better. ?

Despite this, also in view of the completeness of this study, I am still convinced of the significance and potential impact of this work for the field of spintronics and magnonics. Thus, after taking into account this minor remaining point from above I recommend this manuscript for publication in Nature Communications afterwards.

Yours sincerely,

Reviewer

Xin Yu Zheng
Geballe Laboratory for Advanced Materials
McCullough Bldg. Rm 175
476 Lomita Mall
Stanford University
Stanford, CA 94305-4045
TEL: (650) 724-4007
xinzheng@stanford.edu

We thank both referees for considering our responses to their comments. We appreciate that Referee A is satisfied with our response and that Referee B is overall positive of the paper.

We address the remaining minor point by referee B below and describe the associated changes to the manuscript. With these changes, we believe that our manuscript is now ready for publication in *Nature Communications*.

REFEREE B

1. Referee B states that *"I understand that the best data is more favored to be shown but, for instance, the answer to my critics on choosing the film for the current switching experiment is because "for the purpose of current switching of magnetization, not only are thinner films easier to switch, the weaker PMA in the 4.1 nm samples facilitates the SOT switching experiment." as the authors write, is not fully adequate in my opinion.*

The authors stress the fact that indeed the critical switching current density is remarkably low and following the remark of the other reviewer compare it to a YIG/Pt system of 5 nm thickness. First, since there is such strong thickness dependence- and they also say it themselves in the revised version - a true comparison is not possible, the compared values are already for a thicker film. To compare more qualitatively: The authors mention they also measured the critical switching current density for different film thicknesses, do they also have data such as the one where they present the FMR study in the main? Would it be possible to show some dependency of the current density from the film thickness , for instance in the supplemental material? Then the difference in order of magnitude to the YIG/Pt might be also seen even better?"

We now address the referee's request for addressing how the critical current density for SOT switching changes with film thickness. As the referee points out, a comparison across different systems is difficult due to the variations in thickness and anisotropy. We have also performed SOT switching on one other thickness, a 15 nm LAFO film – the same thickness as the one shown in the FMR study in the main manuscript.

More specifically, in thicker films, the anisotropy is much stronger, and therefore a larger current is required to achieve SOT switching. For 15 nm films, the critical switching current is on the order of 10^7 A/cm² for in-plane fields in the few 10s of mT range. These values are similar to those previously reported in Pt/YIG systems. However, we would like to emphasize that LAFO's anisotropy can be tuned dramatically with thickness. In particular, we can achieve very low anisotropy at low thicknesses, which is a combination not present in the garnet systems. Both the low anisotropy and low thickness contribute to lowering the critical switching current, allowing us to achieve record low switching currents at low thicknesses.

As the referee requests, we now show this data in Supplementary Section 7 and Supplemental figure S9. We also add an explanation on page 4 right column last paragraph clarifying the unique combination of low anisotropy and low thickness in LAFO, as well as a referral to Supplement Section 7: “*We also performed SOT switching in a 15 nm LAFO film (Supplemental Materials Section 7), where the J_c is comparable to those reported in Pt/YIG systems due to the larger anisotropy of thicker LAFO films. However, LAFO provides a rare combination of weak PMA at ultra-low thicknesses, both of which help lower J_c . This combination allows the realization of ultra-low J_c at low thicknesses.*”

By addressing the final minor point of referee B and making the appropriate revisions to the manuscript, we believe that our manuscript is ready for publication in *Nature Communications*. We attach below a list of major modifications to the manuscript.

Sincerely,

Xinyu Zheng

Sanyum Channa

Yuri Suzuki

Xin Yu Zheng, Sanyum Channa, and Yuri Suzuki

LIST OF MAJOR CHANGES:

1. **Page 4 right column last paragraph:** “*We also performed SOT switching in a 15 nm LAFO film (Supplemental Materials Section 7), where the J_c is comparable to those reported in Pt/YIG systems due to the larger anisotropy of thicker LAFO films. However, LAFO provides a rare combination of weak PMA at ultra-low thicknesses, both of which help lower J_c . This combination allows the realization of ultra-low J_c at low thicknesses.*”
2. **Supplemental materials section 7:** “*In addition to the SOT switching data on the 4 nm LAFO film shown in the manuscript, we also performed SOT switching on a thicker 15 nm LAFO film as shown in Fig. S9. These thicker films have a much larger anisotropy and therefore a larger current is required for switching. The critical switching current J_c is about 21×10^6 A/cm² and 17.7×10^6 A/cm² for in-plane fields of 10 mT and 50 mT respectively. These values are similar to those reported in YIG systems. However, LAFO*

provides an advantage in that its anisotropy can be lowered for thinner samples, both of which reduce J_c , allowing for record low critical switching currents at low thicknesses.”

3. Added supplemental figure S9 showing SOT switching on a Pt(5 nm)/LAFO(15 nm)/MGO sample.